# A Noninvasive Method of Diagnosing Metabolic Dysfunction-Associated Steatohepatitis Using Cytokeratin-18 Fragment and FIB-3 Index

**DOI:** 10.3390/diagnostics15081023

**Published:** 2025-04-17

**Authors:** Tomoko Tadokoro, Miwa Kawanaka, Hirokazu Takahashi, Shinichi Aishima, Wenli Zhao, Rie Yano, Kei Takuma, Mai Nakahara, Kyoko Oura, Koji Fujita, Kiyoyuki Kobayashi, Shima Mimura, Joji Tani, Asahiro Morishita, Reiji Haba, Tsutomu Masaki, Hideki Kobara, Masafumi Ono

**Affiliations:** 1Department of Gastroenterology and Neurology, Faculty of Medicine, Kagawa University School of Medicine, Kagawa 761-0793, Japan; yano.rie@kagawa-u.ac.jp (R.Y.); takuma.kei@kagawa-u.ac.jp (K.T.); nakahara.mai@kagawa-u.ac.jp (M.N.); oura.kyoko@kagawa-u.ac.jp (K.O.); fujita.koji@kagawa-u.ac.jp (K.F.); kobayashi.kiyoyuki@kagawa-u.ac.jp (K.K.); mimura.shima@kagawa-u.ac.jp (S.M.); tani.joji.kb@kagawa-u.ac.jp (J.T.); morishita.asahiro@kagawa-u.ac.jp (A.M.); masaki.tsutomu@kagawa-u.ac.jp (T.M.); kobara.hideki@kagawa-u.ac.jp (H.K.); ono.masafumi@kagawa-u.ac.jp (M.O.); 2Department of General Internal Medicine2, Kawasaki Medical Center, Okayama 700-8505, Japan; m.kawanaka@med.kawasaki-m.ac.jp; 3Division of Metabolism and Endocrinology, Faculty of Medicine, Saga University, Saga 849-8501, Japan; takahas2@cc.saga-u.ac.jp (H.T.); 21624009@edu.cc.saga-u.ac.jp (W.Z.); 4Liver Center, Saga University Hospital, Faculty of Medicine, Saga University, Saga 849-8501, Japan; 5Department of Scientific Pathology, Graduate School of Medical Sciences, Kyushu University, Fukuoka 812-8582, Japan; aishima.shinichi.476@m.kyushu-u.ac.jp; 6Division of Innovative Medicine for Hepatobiliary and Pancreatology, Faculty of Medicine, Kagawa University School of Medicine, Kagawa 761-0793, Japan; 7Department of Diagnostic Pathology, Faculty of Medicine, Kagawa University School of Medicine, Kagawa 761-0793, Japan; haba.reiji@kagawa-u.ac.jp

**Keywords:** metabolic dysfunction-associated steatotic liver disease, metabolic dysfunction-associated steatohepatitis, cytokeratin-18 fragment, fibrosis-4 index, fibrosis-3 index

## Abstract

**Background/Objectives:** We aim to determine if cytokeratin-18 fragment (CK-18F) could be used to diagnose metabolic dysfunction-associated steatohepatitis (MASH). **Methods:** A total of 289 patients with metabolic dysfunction-associated steatotic liver disease (MASLD) were enrolled in the analysis. To evaluate the association between CK-18F levels and the histological features of MASH, weighted receiver operating characteristic (ROC) curve analyses were performed. The diagnostic utility of CK-18F was compared with that of the Mac-2 binding protein glycan isomer (M2BPGi). Additionally, we assessed the predictive performance of combining CK-18F with either the FIB-4 index or the FIB-3 index for diagnosing MASH and investigated predictors of future progression to cirrhosis. **Results:** CK-18F was more useful for MASH diagnosis than M2BPGi and the FIB-4 index in the multivariate analysis, with a sensitivity of 47% and specificity of 80% at a CK-18F cutoff value of 750 U/L. Because CK-18F decreases with advanced liver fibrosis, the combination of the FIB-4 or FIB-3 index with CK-18F was examined to identify cases with cirrhosis. The combination of the CK-18F level and the FIB-3 index better predicted MASH than the combination of the CK-18F level and the FIB-4 index. The FIB-3 index was the most useful predictor of cirrhosis on imaging five years after diagnosis with F2 or less disease. **Conclusions:** CK-18F is useful for MASH diagnosis, and the diagnostic algorithm combining CK-18F with the FIB-3 index may be more useful than the previously reported MASH diagnostic algorithm that combined it with the FIB-4 index.

## 1. Introduction

Fatty liver is now the leading cause of liver disease in the Asia Pacific region [1,2,3,4]. Advances in viral hepatitis treatment have led to a rapid increase in the prevalence of fatty liver disease, which is expected to increasingly cause liver-related death in the future [5]. Although fatty liver is generally associated with obesity, non-obese fatty liver accounts for approximately 40% of fatty liver cases, and obesity should not be the sole criterion for fatty liver screening [6]. The terms nonalcoholic fatty liver disease (NAFLD) and nonalcoholic steatohepatitis (NASH), which have long been used to classify fatty liver disease, can potentially cause stigma. An international expert panel recently proposed a new term for fatty liver disease—metabolic dysfunction-associated steatotic liver disease (MASLD)—as an alternative to NAFLD [7]. Patients with adiposity and any one of the following cardiometabolic criteria were considered to have MASLD: abnormal body mass index (BMI) or waist circumference, blood glucose level, blood pressure, triglyceride levels, or HDL-C levels. Based on this new definition, metabolic dysfunction-associated steatohepatitis (MASH) has been selected as an alternative to the traditional term NASH [7]. In parallel with MASLD, MASH is also on the rise, with an estimated global prevalence of 5.27% [8]. MASH is a liver disease with a poor prognosis associated with hepatocellular degenerative necrosis, inflammatory cell infiltration, and fibrosis, and some MASH cases progress to cirrhosis and hepatocarcinoma. Diagnosing MASH among patients with MASLD is important, and at present, the gold standard for this is liver histology using a liver biopsy. However, liver biopsy has many limitations apart from its invasiveness, such as diagnostic costs, sampling errors, and inter-observer variability in diagnosis. Therefore, simpler, less invasive, and more accurate diagnostic methods for MASH/MASLD are needed. Attempts to diagnose MASH noninvasively have been made using blood test data and imaging tests. Among these, cytokeratin-18 fragment (CK-18F), an apoptosis marker, has been reported to be useful as a biomarker for the diagnosis of NASH [9,10,11,12]. There have been reports on the usefulness of CK-18F in the diagnosis of NASH; however, there are few reports on its diagnosis using the new definition of MASH/MASLD. Nevertheless, it has been reported that CK-18F measurement alone is of limited value in diagnosing NASH [13]. CK-18F cannot be used as the only screening method, but some reports indicate that it can be useful in combination with other noninvasive methods [14,15]. Tada et al. proposed a two-step diagnostic algorithm combining the fibrosis-4 (FIB-4) index [16] and CK-18F, which have been commonly used in clinical practice as indicators of liver fibrosis in recent years [9].

However, the FIB-4 index includes age in its formula; therefore, the cutoff value may differ depending on age. Recently, several age-independent scoring systems have been developed, and the FIB-3 index has attracted attention [17,18].

This study examined whether CK-18F is useful as a stand-alone serum marker for diagnosis and prognosis prediction, even for the newly defined MASH/MASLD, and whether it is useful in combination with noninvasive scoring systems such as the FIB-4 and FIB-3 indices.

## 2. Materials and Methods

### 2.1. Patients

Between 1990 and 2023, 308 patients with suspected MASH underwent liver biopsies at Kagawa University (*n* = 114), Saga University (*n* = 119), and Kawasaki Medical School (*n* = 75). All eligible patients who met the MASLD criteria [19] during the study period were included. Inclusion criteria were as follows: age ≥ 18 years, fulfillment of the diagnostic criteria for MASLD, and having undergone liver biopsy. Exclusion criteria included the presence of other liver diseases, severe extrahepatic comorbidities, current pregnancy or lactation, coexisting hepatocellular carcinoma or other malignancies, and the absence of stored serum samples. Of these, 289 patients were included in the analysis (Figure 1). Patients with adiposity and any one of the following cardiometabolic criteria were considered to have MASLD: abnormal body mass index (BMI) or waist circumference, blood glucose level, blood pressure, triglyceride level, or HDL-C level. The patient background information included sex, height, weight, BMI, age at diagnosis, comorbidities (abnormal BMI/waist circumference, blood glucose level, blood pressure, triglyceride levels, or HDL-C levels), liver function tests, liver fibrosis markers (Mac-2 binding protein glycan isomer [M2BPGi]), liver disease-related fibrosis scores (FIB-4 index and FIB-3 index), and liver histology results. Because this study was conducted among Asians, overweight or obesity was defined as a BMI of 23 or higher.

### 2.2. Histopathological Review and Diagnostic Criteria for MASH

Formalin-fixed, paraffin-embedded liver sections were stained with hematoxylin and eosin or azan. Digital images of the soft biopsy samples were captured using a batch slide scanner (NanoZoomer 3.2.15; Hamamatsu Photonics KK, Shizuoka, Japan). The diagnosis of MASLD/MASH was initially made by experienced pathologists at each participating institution. To ensure consistency and diagnostic accuracy, a central histopathological review was also performed by a board-certified hepatopathologist with expertise in liver pathology. The NAFLD Activity Score (NAS) [20], which is commonly used to evaluate liver histology in NASH, was used for histological classification. Based on previous reports, MASH was defined as having a NAS of 5 or higher [9,10]. In addition, fibrosis stages were defined according to Brunt’s classification as follows: stage 1, zone 3 perisinusoidal fibrosis; stage 2, portal fibrosis; stage 3, bridging fibrosis; and stage 4, cirrhosis [21]. The pathologist was blinded to the patients’ laboratory data.

### 2.3. CK-18F Level Measurements

Blood samples were taken from all the patients at the time of liver biopsy and stored at −80 °C. Serum CK-18F cleaved by caspases was measured using an IMMUNIS CK-18F enzyme immunoassay (EIA)^®^ (Institute of Immunology, Co., Ltd., Tokyo, Japan) by sandwich EIA.

### 2.4. Liver Disease-Related Fibrosis Score

The FIB-3 and FIB-4 indices [22] were compared. The FIB-3 and FIB-4 indices were calculated using the following formula [16,17].FIB-3 index = 5 × ln AST (IU/L) − 2 × ln ALT (IU/L) − 0.18 × PLT (× 10^4^/μL) − 5FIB-4 index = (Age × AST [IU/L])/(PLT [10^9^/L] × √ALT [IU/L])

### 2.5. Association with Liver Fibrosis Progression

We examined the progression of liver fibrosis after 5 years in patients with a fibrosis stage of 2 or less at the time of liver biopsy. Advanced liver fibrosis was defined as a vibration-controlled transient elastography after 5 years of >7.9 kPa [23] or cirrhotic findings on abdominal ultrasound (irregular or nodular liver surface, blunt hepatic margins, abnormal liver parenchyma, liver morphologic changes, and symptoms of portal hypertension) [24], and its association with CK-18F, FIB-4 index, and FIB-3 index was investigated.

### 2.6. Statistical Analysis

Representative values are presented as medians (interquartile range [IQR]). Categorical variables are presented as numbers and percentages. Comparisons between the two groups were performed using the *t*-test, Mann–Whitney U test, chi-square test, and Fisher’s exact probability test. The Shapiro–Wilk test was used to evaluate the normality of continuous variables. For variables that did not follow a normal distribution, non-parametric statistical methods were employed. Tukey’s test was used as a multiple comparison test. Statistical significance was set at *p* < 0.05. Correlation coefficients were calculated using Pearson’s method to analyze the association of CK-18F with patient background, hematological and biochemical data, and histological images obtained by liver biopsy. The M2BPGi, FIB-3 index, and FIB-4 index were also analyzed in the same manner to compare their usefulness. For each continuous variable, the optimal cutoff value was identified using the Youden index and receiver operating characteristic (ROC) curve analysis. Univariate and multivariate analyses were performed to identify factors potentially useful for distinguishing between MASH and MASLD. In the univariate analysis, each variable was assessed individually to evaluate its association with the differentiation of MASH and MASLD. Variables with significant associations (*p* < 0.05) were included in the multivariate analysis to adjust for potential confounding factors and to identify independent predictors. A decision tree model was then created for MASH diagnosis. All statistical analyses were performed using JMP Pro 17.0 (SAS Institute, Cary, NC, USA).

## 3. Results

### 3.1. Clinical Characteristics

Of 308 patients with suspected MASLD who underwent percutaneous liver biopsy at the study center, 289 who met the definition of MASLD were included in the analysis. The patient background, serology, and pathology results are shown in Table 1. The median age of the patients was 61 (48–68) years, with a male-to-female ratio of 117/172. Complicating metabolic abnormalities included obesity (BMI > 23) in 269 patients, diabetes (145 patients), hypertension (145 patients), and dyslipidemia (189 patients). A total of 140 patients had NAFLD activity scores in the borderline range (3–4 points), and 104 patients had NAFLD activity scores of 5 or more points (MASH).

### 3.2. Association Between Patient’s Underlying Disease and CK-18F

There was no significant association between sex, age, obesity, diabetes, hypertension, or dyslipidemia and CK-18F values (Figure 2). When examining the relationship between various blood biochemical tests, significant correlations were found between aspartate aminotransferase (AST), alanine aminotransferase (ALT), and CK-18F (Figure 3). In contrast, M2BPGi was poorly associated with hepatobiliary enzymes other than AST.

### 3.3. CK-18F Is Useful in the Diagnosis of MASH

In the liver biopsy study population, serum CK-18F, M2BPGi, FIB-4 index, and FIB-3 index were significantly correlated with fibrosis stage. However, CK-18F expression was reduced at the F4 stage (Figure 4). Serum CK-18F levels increased as NAS levels increased; the FIB-3 and FIB-4 indices were associated with the degree of ballooning and NAS, whereas M2BPGi showed no significant correlation with NAS (Table 2, Figure 5). The optimal cutoff value for CK-18F detected based on its correlation with NAS was 750 U/L, with a sensitivity, specificity, positive predictive value, and negative predictive value of 47%, 80%, 57%, and 73%, respectively. Then, the optimal cutoff value for the FIB-3 index detected based on its correlation with NAS was 4.1.

### 3.4. MASH Diagnostic Algorithm Based on CK-18F and the FIB-3 Index

CK-18F is useful to a certain extent for MASH diagnosis; however, for clinical application, we examined the suitability of the algorithm proposed by Tada et al. and created a revised version, that is, a two-step diagnostic algorithm for MASH using CK-18F and the FIB-3 index. According to the latest meta-analysis, the mean cutoff value of CK18-F for predicting MASH and related conditions is 264.3 U/L (range: 74.0–466.0 U/L) [25]; however, in the present cohort, 750 U/L was more accurate for detecting MASH. The cutoff value for the FIB-3 index was 4.1, as described above. The algorithm for MASH diagnosis using CK-18F and the FIB-3 index showed a better correlation with a MASH diagnosis than the algorithm created using CK-18F and the FIB-4 index [9] (Table 3, Figure 6).

### 3.5. Predictors of Liver Fibrosis Progression

Seventy-five patients who had F2 or less disease on liver biopsy and were able to undergo imaging after 5 years were included in the study. Those with advanced fibrosis at 5 years had higher CK-18F, FIB-3 index, and FIB-4 index at diagnosis than those without advanced fibrosis. The FIB-3 index in the ROC curve had an AUROC of 0.82 (95% confidence interval: 1.49–3.26), which was the best value. The cutoff value was 2.86, with a sensitivity, specificity, positive predictive value, and negative predictive value of 88%, 70%, 60%, and 92%, respectively (Figure 7).

## 4. Discussion

CK-18F was more useful than M2BPGi in diagnosing MASLD/MASH in a cohort that met the criteria for MASLD (Table 2). In addition, the new scoring system using the FIB-3 index was as useful as or more useful than the FIB-4 index for diagnosing MASH (Figure 5).

Although liver biopsy remains the gold standard for staging and monitoring MASLD [26], considering the high prevalence of MASLD and the invasiveness and medical costs of liver biopsy, it is not feasible for all patients. It is important for family physicians to select patients who may have advanced liver disease through noninvasive testing and establish a referral process to a gastroenterologist for definitive diagnosis and close follow-up, including liver biopsy [3,4].

The intracellular soluble cytokeratin-18, the major intermediate filament protein of the liver, is released extracellularly during cell death [27]. When apoptosis occurs, CK-18 is fragmented by caspases [28]. Soluble full-length and fragments of CK-18 can be detected in human serum by an enzyme-linked immunoassay (ELISA); the 30 kDa fragment can be detected using a specific antibody (M30) and is said to reflect apoptosis [29]. For detecting apoptosis, caspase activity is transient and variable, but CK-18 accumulates; therefore, these antibodies can be used for more accurate detection of apoptosis. Previous reports have described serum CK-18F levels as a useful biomarker for NASH and fibrosis in patients with NAFLD [9,10,27], with limited usefulness as a single marker [13,15]. Additionally, some reports suggest that it is useful in conjunction with other scoring systems, such as the FIB-4 index [9,14]. However, there are concerns about the diagnostic accuracy of the FIB-4 index in an aging society. This study similarly suggested that the CK-18F value is useful for differentiating MASLD from MASH; however, its usefulness was considered limited when the sensitivity and specificity were examined. This may be because CK-18F levels decrease as hepatic fibrosis progresses, as observed in the F4 stage. Previous studies have questioned the usefulness of CK-18F alone [13]. Therefore, we created a more accurate algorithm using the FIB-3 index instead of the FIB-4 index, referring to the two-step diagnostic algorithm proposed by Tada et al. that combined the FIB-4 index and CK-18F [9] (Figure 6). Figure 6a shows how the current cohort fits into a previous classification based on the algorithm proposed by Tada et al. If we change step 1, that is, the FIB-4 index to the FIB-3 index, the AUC is even better (Figure 6b). Furthermore, when the cutoff value for CK-18F was set to 750 U/L (Figure 6c), the AUC was the best, and the diagnostic performance was high. In the study conducted by Tada et al., NASH was diagnosed based on (1) any degree of steatosis with centrilobular ballooning or both (Mallory–Denk bodies) or (2) any degree of steatosis with centrilobular pericellular/perisinusoidal fibrosis or bridging fibrosis in the absence of another identifiable cause. However, in the study by Tada et al., patients who were classified as having NASH but had a NAS score of 2 or less were considered to have NAFL [9]. Therefore, this study is not directly comparable to the present study, which defined MASH as having a NAS of 5 or higher. Additionally, another study that combined the FIB-4 index with CK-18F focused on identifying at-risk MASH [15], which differs from the target population in the present study. These differences may have contributed to the variation in cutoff values and other parameters. While the two-step algorithm remains useful across studies, the present study was conducted using a broader and more general definition.

Therefore, the new scoring system, the FIB-3 index combined with CK-18F, had higher discriminative power for MASH than the old system. CK-18F is a marker of hepatocyte apoptosis and may not adequately reflect fibrosis, although it is closely associated with ballooning and other factors. However, when used in conjunction with the new liver fibrosis scoring system, the FIB-3 index, the accuracy of detecting MASH that requires close examination may be improved. The FIB-3 index can be calculated from AST, ALT, and platelets; therefore, it can be used in general clinical practice. The addition of the CK-18F test to the tests performed for patients with an elevated FIB-3 index may be more cost-effective and accurate for the diagnosis of MASH.

Interestingly, the FIB-3 index was also useful for predicting the progression of liver fibrosis (Figure 7). Fibrosis is significantly associated with life expectancy in MASLD [26]. In contrast, CK-18F correlated with AST and ALT (Figure 3), but its correlation with fibrosis stage was rather weak, especially when fibrosis progressed to F4 (Figure 4). This result indicates that CK-18F reflects apoptosis, but another index is required to assess fibrosis. The FIB-4 index formula includes age, which can result in an oversight in younger patients with MASLD. The FIB-3 index, which solves this problem, may be an important scoring system for liver fibrosis.

This study had several limitations. The patient population analyzed was limited to university hospitals and may have attributes different from those of the general population. There were no clear criteria for performing a liver biopsy, which were left to the judgment of each physician. Because only cases in which liver biopsies were performed were included in the analysis, it is possible that a population with a particularly severe degree of liver damage was selected. Perhaps because of this, the area under the curve for the diagnosis of CK-18F, FIB-4 index, and FIB-3 index was lower in this study than in previous reports. The cutoff value for CK-18F was also higher than that previously reported, but this may also be due to the presence of many patients with severe hepatic impairment. In the future, to determine optimal and more generalizable cutoff values for clinical application, it will be important to conduct prospective multicenter studies that include regional core hospitals and general clinics. In addition, cases analyzed for long-term prognosis may require some form of therapeutic intervention, and the diagnosis of cirrhosis by imaging may differ from that by pathologic examination. In addition, sampling errors in the baseline liver biopsy may have affected the results. However, few studies have compared CK-18F and M2BPGi levels and examined the predictors of liver fibrosis progression in MASH, which we considered clinically valuable.

## 5. Conclusions

CK-18F has the unique feature of reflecting hepatocyte apoptosis and may be a useful biomarker for the diagnosis of MASH. Combining serum CK-18F values with the FIB-3 index will achieve a highly accurate diagnosis of MASLD, which requires specialist care.

## Figures and Tables

**Figure 1 diagnostics-15-01023-f001:**
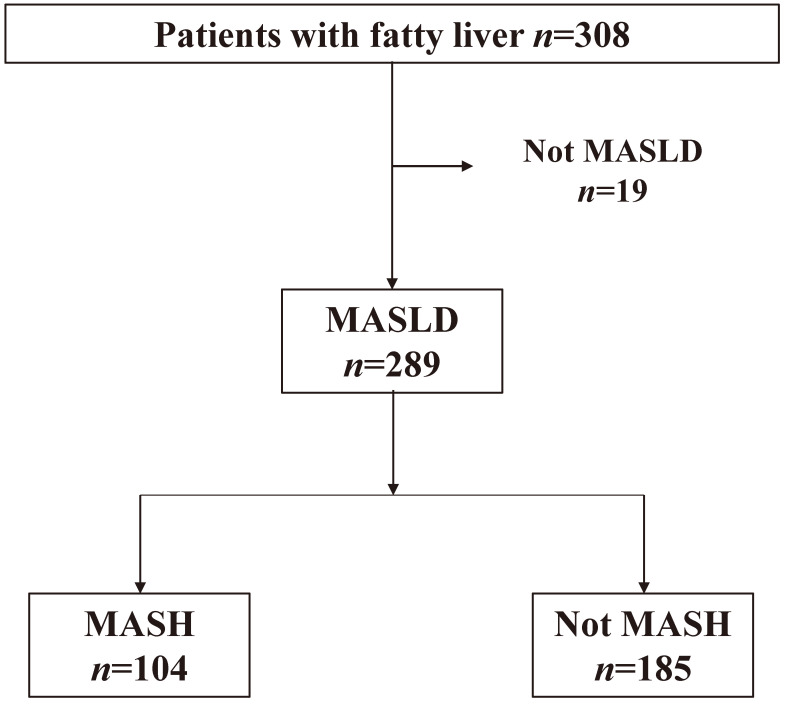
Patient Selection Flowchart for MASLD Diagnosis.

**Figure 2 diagnostics-15-01023-f002:**
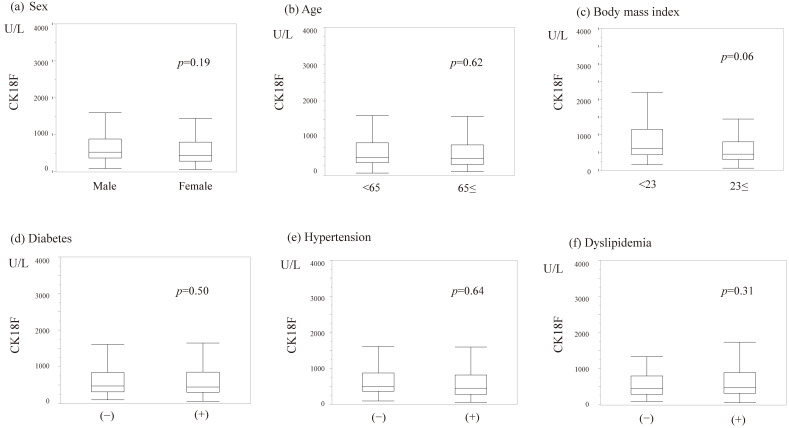
Patients’ basic condition and CK-18F. (**a**) Sex and CK-18F; (**b**) Age and CK-18F; (**c**) Body mass index and CK-18F; (**d**) Diabetes and CK-18F; (**e**) Hypertension and CK-18F; and (**f**) Dyslipidemia and CK-18F. Comparisons with binary categorical variables were performed using the Mann–Whitney U test. There was no significant association between sex, age, obesity, diabetes, hypertension, or dyslipidemia and CK-18F values. CK-18F, cytokeratin-18 fragment.

**Figure 3 diagnostics-15-01023-f003:**
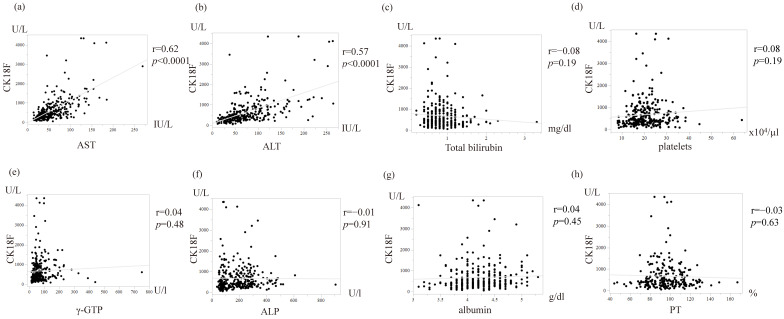
Blood biochemistry data and CK-18F. (**a**) AST and CK-18F; (**b**) ALT and CK-18F; (**c**) Total bilirubin and CK-18F; (**d**) Platelets and CK-18F; (**e**) γ-GTP and CK-18F; (**f**) ALP and CK-18F; (**g**) Albumin and CK-18F; and (**h**) PT and CK-18F. Significant correlations were observed between AST, ALT, and CK-18F (*p* < 0.0001). Pearson’s correlation coefficients were calculated to examine the relationship between CK-18F levels and blood test parameters. CK-18F, cytokeratin-18 fragment; AST, aspartate aminotransferase; ALT, alanine aminotransferase; γ-GTP, gamma-glutamyl transpeptidase; ALP, alkaline phosphatase; PT, prothrombin time.

**Figure 4 diagnostics-15-01023-f004:**
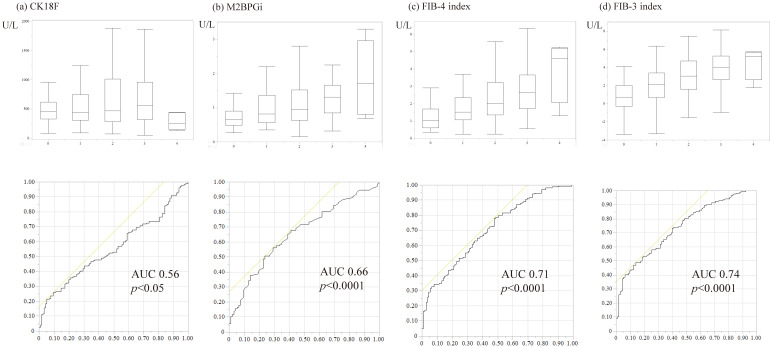
Association of fibrosis stage on pathological diagnosis with biomarkers and liver fibrosis scores. (**a**) Cytokeratin-18 fragment; (**b**) Mac-2 binding protein glycan isomer; (**c**) Fibrosis-4 index; (**d**) Fibrosis-3 index. The stages of fibrosis were as follows: stage 1, zone 3, perisinusoidal fibrosis; stage 2, as above with portal fibrosis; stage 3, as above with bridging fibrosis; and stage 4, cirrhosis. In the liver biopsy study population, serum CK-18F, M2BPGi, FIB-4 index, and FIB-3 index were significantly correlated with fibrosis stage (*p* < 0.05). CK-18F expression is reduced at the F4 stage. The ROC curve analysis was performed for each continuous variable. The AUC of the discriminative power of MASH (NAS score of 5 or higher) is described. The yellow line represents the reference line for random prediction. AUC, area under the curve; CK-18F, cytokeratin-18 fragment; M2BPGi, Mac-2 binding protein glycan isomer; FIB-4 index, fibrosis-4 index; FIB-3 index, fibrosis-3 index.

**Figure 5 diagnostics-15-01023-f005:**
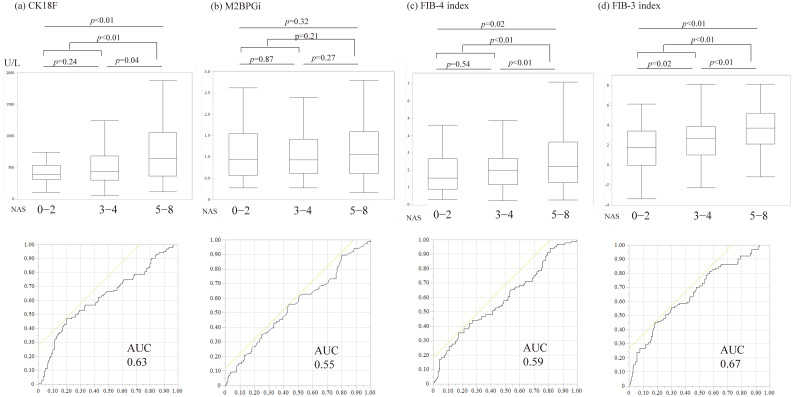
Correlation of NAFLD activity score with biomarkers or liver fibrosis scores. (**a**) Cytokeratin-18 fragment; (**b**) Mac-2 binding protein glycan isomer; (**c**) Fibrosis-4 index; and (**d**) Fibrosis-3 index. Comparisons with binary categorical variables were performed using the Mann–Whitney U test. The ROC curve analysis was performed for each continuous variable. Serum CK-18F levels increased as NAS levels increased; the FIB-3 and FIB-4 indices were also associated with the degree of ballooning and NAS, while M2BPGi showed no significant correlation with NAS. The yellow line represents the reference line for random prediction. CK-18F, cytokeratin-18 fragment; M2BPGi, Mac-2 binding protein glycan isomer; FIB-4 index, fibrosis-4 index; FIB-3 index, fibrosis-3 index; NAS, nonalcoholic fatty liver disease activity score.

**Figure 6 diagnostics-15-01023-f006:**
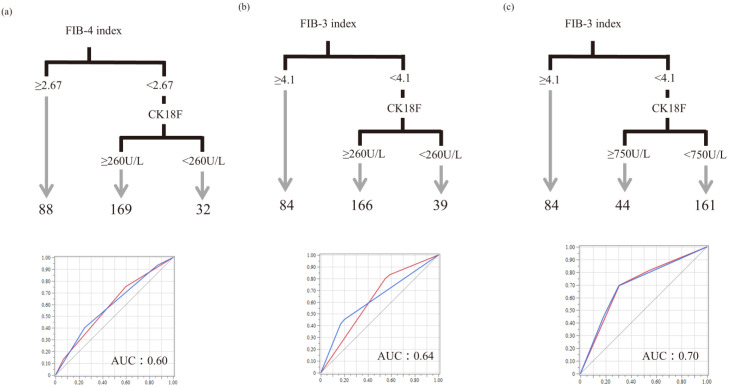
Two-step diagnostic algorithm for MASH using CK-18F and the FIB-3 index. A decision tree model was then created for MASH diagnosis. The current cohort was fitted into a previous classification based on an algorithm proposed by Tada et al. (**a**). The AUC was even better when the FIB-4 index was replaced with the FIB-3 index (**b**). Furthermore, a cutoff value of 750 U/L for CK-18F (**c**) yielded the best AUC and a high diagnostic performance. The two-step method using the FIB-3 index and CK-18F (**b**) provides a more accurate diagnosis than the combination of the FIB-4 index and CK-18F (**a**). At a CK-18F cutoff value of 750 U/L (**c**), the AUC was higher than that of the previously reported cutoff value of 260 U/L 9 (**b**). Red line represents NAS 0-4, blue line represents NAS 5 and above. CK-18F, cytokeratin-18 fragment; FIB-4 index, fibrosis-4 index; FIB-3 index, fibrosis-3 index; AUC, area under the curve.

**Figure 7 diagnostics-15-01023-f007:**
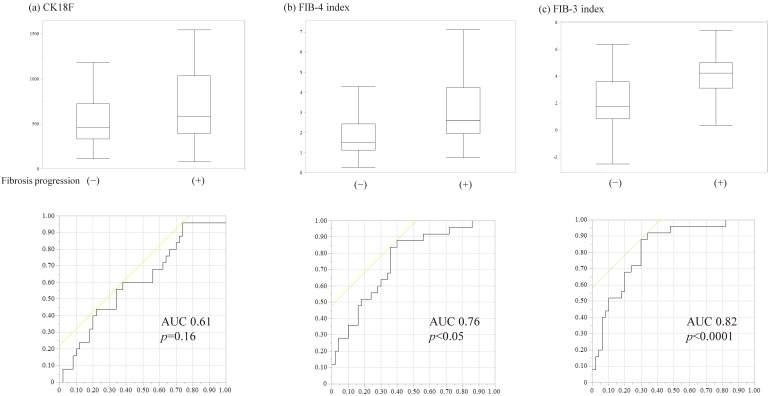
Predictors of liver fibrosis progression at 5 years (fibrosis stage 0–2 at diagnosis *n* = 75). (**a**) Cytokeratin-18 fragment, (**b**) Fibrosis-4 index, and (**c**) Fibrosis-3 index. The FIB-3 index was useful for predicting future liver fibrosis progression. Comparisons with binary categorical variables were performed using the Mann–Whitney U test. ROC curve analysis was performed to evaluate the association between each variable and the future development of liver cirrhosis. The yellow line represents the reference line for random prediction. CK-18F, cytokeratin-18 fragment; FIB-4 index, fibrosis-4 index; FIB-3 index, fibrosis-3 index; AUC, area under the curve.

**Table 1 diagnostics-15-01023-t001:** Clinical and histological characteristics of patients with metabolic dysfunction-associated steatotic liver disease.

Characteristic *n* = 289	
Age (years), median (IQR)	61 (48–68)
Sex, males	117 (40.5%)
BMI (kg/m^2^) median (IQR)	27.9 (25.5–31.0)
ALT (IU/L), median (IQR)	66 (43–100)
AST (IU/L), median (IQR)	51 (36–77)
gamma-GTP (IU/L), median (IQR)	57 (41–84)
Total bilirubin (mg/dL), median (IQR)	0.8 (0.6–1.0)
Total cholesterol (ng/dL), median (IQR)	191 (167–214)
Platelet count (10^4^/µg), median (IQR)	19.5 (16.1–24)
Ferritin (ng/dL), median (IQR)	214 (104–346)
Type 4 collagen 7S (ng/mL), median (IQR)	4.9 (3.8–6.4)
Hyaluronic acid (ng/mL), median (IQR)	52 (23–104)
Cytokeratin-18 (U/L), median (IQR)	463 (310–844)
M2BPGi, median (IQR)	1 (1–2)
FIB-4 index, median (IQR)	2 (1–3)
FIB-3 index, median (IQR)	3 (1–4)
NAFLD activity score, 0–2/3–4/5–8 (*n*)	45/140/104
Ballooning, 0/1/2 (*n*)	56/151/82
Steatosis, 0/1/2/3 (*n*)	13/183/65/28
Fibrosis, 0/1/2/3/4 (*n*)	34/81/94/75/5
Lobular inflammation, 0/1/2/3 (*n*)	7/163/92/27

BMI: body mass index; ALT: alanine aminotransferase; AST: aspartate aminotransferase; gamma-GTP: gamma-glutamyl transpeptidase; M2BPGi: Mac-2 binding protein glycan isomer; FIB-4 index: Fibrosis-4 index; FIB-3 index: Fibrosis-3 index; IQR: Interquartile Range.

**Table 2 diagnostics-15-01023-t002:** Factors predicting MASH/MASLD.

Characteristic	MASLD (*n* = 185)	MASH (*n* = 104)	Univariate Analysis			Multivariate Analysis		
			Odds Ratio	95%CI	*p*	Odds Ratio	95%CI	*p*
female	108	64	1.14	0.70–1.86	0.60			
elderly (≥65)	68	37	0.95	0.57–1.57	0.84			
obesity	170	99	1.74	0.65–5.50	0.28			
diabetes	87	58	1.42	0.88–2.30	0.15			
hypertension	92	53	1.05	0.65–1.70	0.84			
dyslipidemia	115	74	1.50	0.89–2.52	0.12			
CK18F (≥750)	37	47	3.30	1.95–5.59	<0.01	2.59	1.48–4.52	<0.01
M2BPGi (≥3.00)	5	6	4.00	0.44–36.04	0.21			
FIB-4 index (≥2.67)	46	42	2.05	1.22–3.42	<0.01	1.36	0.75–2.47	0.31
FIB-3index (≥1.892)	103	83	3.15	1.80–5.51	<0.01	2.04	1.48–3.95	0.03

Univariate and multivariate analyses were performed to identify factors potentially useful for distinguishing between MASH and MASLD. In the univariate analysis, each variable was assessed individually to evaluate its association with the differentiation of MASH and MASLD. Variables with significant associations (*p* < 0.05) were included in the multivariate analysis to adjust for potential confounding factors and to identify independent predictors. MASLD, metabolic dysfunction-associated steatotic liver disease; MASH, metabolic dysfunction-associated steatohepatitis;CK18F, cytokeratin-18 fragment; M2BPGi, Mac-2 binding protein glycan isomer; CI, Confidence interval.

**Table 3 diagnostics-15-01023-t003:** FIB-3 index and CK18F in MASH/MASLD cases.

		CK18F	
		<750 U/L	≥750 U/L
MASH			
FIB-3 index	<4.1	34	23
	≥4.1	23	24
MASLD			
FIB-3 index	<4.1	127	21
	≥4.1	21	16

MASLD, metabolic dysfunction-associated steatotic liver disease; MASH, metabolic dysfunction-associated steatohepatitis; CK18F, cytokeratin-18 fragment; FIB-3 index, fibrosis-3 index.

## Data Availability

The data supporting the findings of this study are available from the corresponding author upon reasonable request.

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
