# Peer review of "A Noninvasive Method of Diagnosing Metabolic Dysfunction-Associated Steatohepatitis Using Cytokeratin-18 Fragment and FIB-3 Index"

_diagnostics, 2025, doi:10.3390/diagnostics15081023_

Round 1

Reviewer 1 Report

Comments and Suggestions for Authors

Dear Authors,

Thanks for the interesting article, you have written a similar article as:

“Serum Cytokeratin 18 Fragment Is an Indicator for Treating Metabolic Dysfunction-Associated Steatotic Liver Disease, Miwa Kawanaka,1 Yoshihiro Kamada,2 Hirokazu Takahashi,3,4 Michihiro Iwaki,5 Ken Nishino,1 Wenli Zhao,3 Yuya Seko,6 Masato Yoneda,5 Yoshihito Kubotsu,3 Hideki Fujii,7 Yoshio Sumida,8 Hirofumi Kawamoto,1 Yoshito Itoh,6 Atsushi Nakajima,5 and the Japan Study Group of NAFLD (JSG-NAFLD), Gastro Hep Advances 2024;3:1120–1128”.What are results differences between two articles? Your aim was” This study examined whether CK-18F is useful as a stand-alone serum marker for diagnosis and prognosis prediction, even for the newly defined MASH/MASLD, and whether it is useful in combination with noninvasive scoring systems such as the FIB-4 and FIB-3 indices” You have studied 3 above factors in the previous study. Please explain the gap in knowledge and the novelty of your study.

Please add a method section to the abstract.

The background section of the abstract is long.

Please mention the method of sample size determination.

Please add inclusion and exclusion criteria.

Please add the method of normality determination to statistical methods.

Please add statistical methods to Figures and tables.

Please add the ethical number

Regards,

Author Response

We would like to thank you very much for reviewing our manuscript and providing valuable comments. We have carefully addressed all the points raised by the reviewers. Responses to Reviewer 1's comments are highlighted in yellow, and responses to Reviewer 2's comments are highlighted in blue.

For Reviewer1

Thanks for the interesting article, you have written a similar article as:

“Serum Cytokeratin 18 Fragment Is an Indicator for Treating Metabolic Dysfunction-Associated Steatotic Liver Disease, Miwa Kawanaka,1 Yoshihiro Kamada,2 Hirokazu Takahashi,3,4 Michihiro Iwaki,5 Ken Nishino,1 Wenli Zhao,3 Yuya Seko,6 Masato Yoneda,5 Yoshihito Kubotsu,3 Hideki Fujii,7 Yoshio Sumida,8 Hirofumi Kawamoto,1 Yoshito Itoh,6 Atsushi Nakajima,5 and the Japan Study Group of NAFLD (JSG-NAFLD), Gastro Hep Advances 2024;3:1120–1128”. What are results differences between two articles? Your aim was” This study examined whether CK-18F is useful as a stand-alone serum marker for diagnosis and prognosis prediction, even for the newly defined MASH/MASLD, and whether it is useful in combination with noninvasive scoring systems such as the FIB-4 and FIB-3 indices” You have studied 3 above factors in the previous study. Please explain the gap in knowledge and the novelty of your study.

→Thank you for your valuable comments. As you pointed out, the previous study examined the relationship between the FIB-4 index and CK18F. In the present study, we expanded the number of participating centers and adopted a different classification, distinguishing between MASH and at-risk MASH.

Although the combination of FIB-4 index and CK18F was also useful in diagnosing MASH, we have some concerns regarding the diagnostic accuracy of the FIB-4 index in the context of an aging society. Therefore, we focused on the recently proposed FIB-3 index, which was found to further improve the diagnostic accuracy for MASH.

This screening approach may allow us to identify MASH in younger individuals, who may otherwise go undetected. We believe this represents one of the novel aspects of our study. However, the cutoff value of CK18F was not consistent between the two studies. This discrepancy may be due to differences in the definition of MASH and variations between institutions. As CK18F is a novel biomarker, further investigation is needed to determine its appropriate cutoff value. We have added a statement regarding this point in the discussion section.

Please add a method section to the abstract.

The background section of the abstract is long.

→Thank you for your comments. I assumed that the difficulty in distinguishing between the Background/Objectives and Methods sections was the cause of the issue. Based on your suggestion, I have revised the abstract accordingly.

Please mention the method of sample size determination.

→Thank you for your valuable comment. The sample size in this study was not determined based on a formal statistical calculation. Instead, all eligible patients who met the inclusion criteria during the study period were included. We have added a statement to clarify this point in the Methods section.

Please add inclusion and exclusion criteria.

→Thank you for your valuable comment. Inclusion criteria were as follows: age ≥18 years, fulfillment of the diagnostic criteria for MASLD, and having undergone liver biopsy.

Exclusion criteria included the presence of other liver diseases, severe extrahepatic comorbidities, current pregnancy or lactation, coexisting hepatocellular carcinoma or other malignancies, and absence of stored serum samples. We have added a description of this information in the Methods section.

Please add the method of normality determination to statistical methods.

→Thank you for your comment. We have added a description of the method used to assess data normality in the Statistical Analysis section. Specifically, the Shapiro–Wilk test was used to evaluate the normality of continuous variables. For variables that did not follow a normal distribution, non-parametric statistical methods were employed.

Please add statistical methods to Figures and tables.

→Thank you for your comments. Following your advice, I have added statistical explanations to the figures and tables.

Please add the ethical number

→Thank you for your comment. The ethical approval number is provided in the Institutional Review Board Statement.

Reviewer 2 Report

Comments and Suggestions for Authors

Fatty liver disease has become the primary cause of liver disease in the Asia-Pacific region. This study aimed to determine whether ‌cytokeratin 18 fragment (CK-18F)‌ can be used to diagnose MASH and to compare its diagnostic performance with ‌Mac-2 binding protein glycosylation isomer (M2BPGi)‌ and fibrosis indices (‌FIB-4‌ and ‌FIB-3‌) in a cohort from three Japanese university hospitals. The strength is that Liver biopsy was used to confirm MASLD/MASH diagnosis, with the ‌NAFLD Activity Score (NAS)‌ system applied.  The combination of CK-18F with FIB-3 showed the best diagnostic performance for MASH, surpassing CK-18F combined with FIB-4.

  1. ‌The study population was recruited from three university hospitals (tertiary care centers), introducing potential selection bias. This cohort may not represent the general population or patients with mild fatty liver disease. The cut-off values of CK-18 should be cited by pooled results from meta-analysis such as Ye J, Lai J, Luo L, Zhou T, Sun Y, Zhong B. Cytokeratin 18 fragment in liver inflammation and fibrosis: Systematic review and meta-analysis. Clin Chim Acta. 2025 Mar 1;569:120147. doi: 10.1016/j.cca.2025.120147. Epub 2025 Jan 19. PMID: 39832704.
  2. Patients in tertiary hospitals often have more advanced or complex conditions, which might overestimate the diagnostic performance of biomarkers like CK-18F.
  3. Patient Selection Flowchart for MASLD Diagnosis is two simple, please provide a version in detail, especially for those excluded by various reasons.
  4. Histopathological Review should be confirmed by another experienced histopathologist.
  5. Why not use CK-18F as the initial screening tests? Two-step diagnostic algorithm for MASH using CK-18F and the FIB-3 index presented a not very good diagnostic performance.

Author Response

We would like to thank you very much for reviewing our manuscript and providing valuable comments. We have carefully addressed all the points raised by the reviewers. Responses to Reviewer 1's comments are highlighted in yellow, and responses to Reviewer 2's comments are highlighted in blue.

For Reviewer2

1‌. The study population was recruited from three university hospitals (tertiary care centers), introducing potential selection bias. This cohort may not represent the general population or patients with mild fatty liver disease. The cut-off values of CK-18 should be cited by pooled results from meta-analysis such as Ye J, Lai J, Luo L, Zhou T, Sun Y, Zhong B. Cytokeratin 18 fragment in liver inflammation and fibrosis: Systematic review and meta-analysis. Clin Chim Acta. 2025 Mar 1;569:120147. doi: 10.1016/j.cca.2025.120147. Epub 2025 Jan 19. PMID: 39832704.

2. Patients in tertiary hospitals often have more advanced or complex conditions, which might overestimate the diagnostic performance of biomarkers like CK-18F.

 →Thank you very much for your valuable comments. We acknowledge that this is an important limitation of our study. Since the analysis was mainly based on patients from a university hospital, we cannot rule out the possibility that the characteristics may differ from those of the general population. However, by combining our approach with the FIB-3 index, we believe we were able to develop an algorithm with broader applicability. Regarding the cutoff value, we have cited the most recent literature you kindly referred to. In accordance with your suggestions, we have incorporated additional descriptions into the Results and Discussion sections.

3.Patient Selection Flowchart for MASLD Diagnosis is two simple, please provide a version in detail, especially for those excluded by various reasons.

→Thank you for your valuable comment. We recognize the importance of providing detailed information regarding patient selection. However, as this was a retrospective study that included liver biopsy cases suspected of MASLD, the only excluded cases were those that did not meet the definition of MASLD as described in the Methods section.

4. Histopathological Review should be confirmed by another experienced histopathologist.

→Thank you for your valuable comment. In this study, all liver biopsy specimens were initially evaluated by experienced pathologists at each participating institution. To ensure consistency and diagnostic accuracy, a central histopathological review was also conducted by a board-certified hepatopathologist with expertise in liver pathology. We believe this approach provided sufficient reliability for the histological assessment. This information has been added to the Methods section.

5. Why not use CK-18F as the initial screening tests? Two-step diagnostic algorithm for MASH using CK-18F and the FIB-3 index presented a not very good diagnostic performance.

→Thank you for your insightful comment. As you pointed out, the diagnostic performance of the two-step algorithm using CK-18F followed by the FIB-3 index was not outstanding when compared to some previous models. However, this diagnostic sequence was not selected arbitrarily; rather, it was automatically generated through decision tree analysis, which identified this combination as the most efficient path based on our dataset. We believe this data-driven approach reflects the real-world applicability of the algorithm within the studied population. Nonetheless, we acknowledge the need for further refinement and external validation of this algorithm in future studies.

Round 2

Reviewer 1 Report

Comments and Suggestions for Authors

Dear Authors,

Thanks for your try in doing recommendations.

Regards,